# Cambial Age Influences PCD Gene Expression during Xylem Development and Heartwood Formation

**DOI:** 10.3390/plants12234072

**Published:** 2023-12-04

**Authors:** Yulia L. Moshchenskaya, Natalia A. Galibina, Tatiana V. Tarelkina, Ksenia M. Nikerova, Maksim A. Korzhenevsky, Ludmila I. Semenova

**Affiliations:** Forest Research Institute, Karelian Research Centre of the Russian Academy of Sciences, 11 Pushkinskaya St., 185910 Petrozavodsk, Russiakarelt@mail.ru (T.V.T.); knikerova@yandex.ru (K.M.N.); maksim.korjan@gmail.com (M.A.K.); mi7enova@gmail.com (L.I.S.)

**Keywords:** *Pinus sylvestris* L., programmed cell death, xylem formation, heartwood, cambial age, gene expression

## Abstract

Heartwood formation is an important ontogenetic stage in Scots pine (*Pinus sylvestris* L.). The amount of heartwood determines the proportion of functionally active sapwood in the total trunk biomass as well as the quality of wood. The key criterion for heartwood formation is the death of xylem ray parenchyma cells. Previously, models that described the patterns of heartwood formation, depending on the cambial age, were derived from Scots pine trees of different ages. The cambial age is the number of annual xylem layers at the core sampling site at a certain trunk height. We studied the features of the occurrence of programmed cell death (PCD) processes during the xylem differentiation and heartwood formation of 80-year-old Scots pine trees, depending on the cambial age, under the lingonberry pine forest conditions in the transition area of the northern taiga subzone and tundra. We have shown that the distance from the cambial zone to the heartwood boundary does not change significantly with stem height. As the cambial age increases, the lifespan of the formed xylem ray parenchyma cells increases and the activity of PCD genes decreases during the formation of both (1) xylem (in the outer layers of sapwood) and (2) heartwood (in the inner layers of sapwood and transition zone). We hypothesized that the decisive factor in the PCD initiation during heartwood formation is the distance of the xylem ray parenchyma cells from the cambial zone. The younger cambium forms wider annual increments, and therefore the xylem ray parenchyma cells in these parts of the trunk reach the distance from the cambial zone earlier, which is necessary for PCD initiation.

## 1. Introduction

*Pinus sylvestris* L. is a heartwood forming tree species. Heartwood is an important factor that determines the quality of wood, and understanding the mechanisms of heartwood formation could potentially allow for controlling this process. Heartwood formation is a form of xylem aging, which is accompanied by various metabolic changes in ray parenchyma cells in the transition zone of sapwood and heartwood [1]. The cambium, known as the lateral meristem, determines xylem formation during ontogenesis. Xylem and phloem precursor cells are produced by pluripotent cells in the cambium. The cambial zone is formed by the cambium, mother cells of the phloem and xylem, and their undifferentiated daughter cells. The cambial zone width is variable and depends on the growing season, while the width of the cambium is usually constant and comprises only a single layer of cells [2]. During the growing season, cambium cell divisions gradually slow down, and the mother xylem cells undergo differentiation. The xylem of conifer plants comprises conductive tracheids and parenchyma cells, forming rays and lining resin ducts. Differentiation of tracheids includes cell extensional growth, secondary cell wall deposition, and programmed cell death (PCD). At the beginning of the elongation stage, the differentiation of xylem cells begins, which is regulated by the transcription factor VASCULAR-RELATED NAC-DOMAIN (VND). Through MYB (myeloblastosis) transcription factors, VND triggers the process of secondary cell wall formation [3]. The synthesis and deposition of secondary cell wall components induces a decrease in cytoplasmic density, a change in tonoplast permeability, which leads to the enlargement and collapse of the vacuole [4]. Soon after the disintegration of the vacuole, degradation of nuclei and DNA occurs in the xylem vessels. It has been shown that the *VND* genes (*VND6* and *VND7*), in addition to regulating the formation of the secondary cell wall, regulate PCD. Many enzymes that degrade cellular contents are direct targets of VND6 and VND7 [5,6,7]. Bifunctional endonuclease (BFN), which has both RNase and DNase activity [8], is a key enzyme responsible for nuclear degradation in xylem vessels and plays a major role during aging and programmed cell death [7]. Important participants in PCD are also cysteine peptidases (CEPs) and metacaspases (MCs), which are involved in the autolysis of cellular contents [9]. After VND activates the expression of genes encoding proteolytic enzymes, BFN and CEP accumulate in the vacuole, and MCs are located in the cytoplasm. With the destruction of the vacuole, enzymes enter the cytoplasm, where autolysis of the cellular contents and part of the cell wall begins [4]. This completes the process of PCD and the formation of tracheal xylem elements. In contrast to tracheids, parenchyma cells lack a clearly defined secondary cell wall. These cells remain viable for many years and take part in various biological processes as well as being a reserve storage of metabolites necessary for lignification of neighboring xylem conducting elements [10]. As one moves toward the center of the trunk, the PCD of xylem ray parenchyma cells occurs, which is a fundamental stage in heartwood formation. The PCD of xylem ray parenchyma cells occurs under molecular genetic control, as well. Previously, *P. sylvestris* showed a significant increase in the activity of genes of the *BFN* family in the transition zone compared to the inner layers of sapwood [11,12].

Previous studies have shown that both height and cambial age can have significant effects on wood formation. The influence of cambial age on the width of annual growth, earlywood and latewood vessel lumen diameter, vessel element length, and earlywood pit membrane diameter was shown [13]. Previously, the patterns of heartwood formation depending on tree age and cambial age within the same tree were studied in the Northwest of Russia in Scots pine trees [4]. We found a very high correlation degree between cambial age and the number of heartwood rings. Therefore, heartwood formation is inextricably linked with cambial aging [14,15].

In this study, we studied changes in the expression profiles of PCD genes in trunk sections with different cambium ages in 80-year-old Scots pine trees.

## 2. Results

### 2.1. The Lifespan of Radial Parenchyma Cells Depends on the Cambial Age

We have shown that the intensity of the PCD processes during xylem differentiation depends on the cambial age. Along with conducting elements, the xylem contains poorly differentiated xylem ray parenchyma cells, which remain viable until the transition from sapwood to heartwood. We compared the lifespan of xylem ray parenchyma cells in *P. sylvestris* stem regions with different cambial ages. Since the transition between sapwood and heartwood is accompanied by the PCD of living ray parenchyma, the number of tree rings with living parenchyma cells was recorded as the sapwood lifespan.

We showed a decrease in the lifespan of xylem ray parenchyma cells from the trunk base to the trunk top (Figure 1A). A strong positive correlation was shown between the lifespan of xylem ray parenchyma cells and the cambial age (Pearson correlation coefficient = 0.96, *p* < 0.01), while the heartwood area percentage of total stem area did not change with stem height or depending on cambial age (Figure 1B). The distance from the cambial zone to the heartwood boundary (width of sapwood plus transition) did not change significantly up to a trunk height of 5 m (Figure 1C). Several samples of ANOVA tests showed significant differences in this indicator for heights of 6 and 7 m (cambial age 26 and 21 years) relative to 0.3 and 1 m (cambial age 79 and 65 years). The differences between other heights were insignificant.

### 2.2. Gene Expression of PCD Genes in Cambial Dormancy and Cambial Activity Period

The onset and duration of cambial activity depends on the temperature. In our study, sampling occurred during the cambial activity period (thickening of the secondary cell wall of early tracheids) (28 June–3 July 2022) and the cambial dormancy period (7–10 October 2021). We investigated the level of expression of genes involved in PCD in the trunk tissues of 80-year-old *P. sylvestris* trees along the radial vector: outer layers of sapwood (differentiating xylem, current year xylem), inner layers of sapwood (two annual rings before transition zone), and transition zone during the period of the cambium dormancy and xylem formation during the period of cambium activity under the lingonberry pine forest conditions in the transition area of the northern taiga subzone and tundra (Pasvik Nature Reserve, Murmansk region, Pechenga district).

In the cambial dormancy period, the expression was shown for *BFN*, *BFN1*, *BFN2* genes. The expression of the remaining genes studied (*BFN3*, *CEP*, *MC5*) was close to zero (outer sapwood) or equal (inner sapwood and transition zone) to zero. The highest level of expression of PCD genes in the cambial dormancy period was shown for the transition zone. During the cambial activity period in outer sapwood, *BFN* gene expression decreased against the background of increased expression of *BFN3*, *CEP*, *MC5* genes, in the inner sapwood the level of *BFN2* expression decreased, and in the transition zone there was a sharp increase (seven times) in the level of *BFN* gene expression (Figure 2).

### 2.3. Principal Component Analysis Using PCD Gene Expression

Principal component analysis was carried out in a data set on the expression levels of the *BFN* family genes (*BFN*, *BFN1*, *BFN2*) in trunk tissues (outer sapwood, inner sapwood, transition zone) of 80-year-old *P. sylvestris* trees sampled during the cambial activity period. Data on the expression of the *BFN3*, *CEP*, *MC5* genes were not used for the analysis, because these genes were not expressed in the inner sapwood and transition zone. For the analysis, we collected gene expression data obtained at different trunk heights (0.3, 1, 2, 3, 4, 5, 6, 7 m) of five model trees.

Principal component analysis of data from 120 models revealed that two factors accounted for 97% of the variability in the item set. The data showed a clear division into three groups according to tissue type (outer sapwood, inner sapwood, transition zone) (Figure 3). Outer sapwood, inner sapwood, transition zone varied by a component of 1, covered 92% of the variance, and was positively correlated with *BFN* gene expression. We can thus conclude that in our study, tissue specificity is the main factor determining the expression profile of PCD genes. We will consider the influence of tissue selection height and cambial age on each tissue separately.

### 2.4. The PCD Process during Xylem Formation during the Cambial Activity Period Depends on the Cambial Age

We studied the expression of PCD genes during the differentiation of xylem cells depending on the cambial age. For this purpose, tissues were collected at different trunk heights—from 0.3 m (cambial age 79 years) to 7 m (cambial age 21 years) from five models of Scots pine trees.

The highest expression levels were shown for the *BFN3*, *CEP*, *MC5* genes. As a result of the analysis, we also showed that sections of the trunk of *P. sylvestris* trees with a younger cambium have higher levels of expression of PCD genes in outer sapwood compared to the older cambium. The strongest negative correlation of the expression level in outer sapwood with the cambial age was shown for the *BFN3* gene (Pearson correlation coefficient, r = −0.70). The average correlation is shown for the *BFN* genes (Pearson correlation coefficient, r = 0.57), *BFN1* (r = 0.60), *CEP* (r = 0.55), and is weak for the *MC5* gene (r = 0.44). *BFN2* gene expression level did not correlate with cambial age (Figure 4).

ANOSIM analysis shows that the expression level of PCD genes in the outer sapwood differed significantly between the sections on different trunk heights (ANOSIM with Bray−Curtis similarity; global R of 0.7568; significance level 0.0001) (Figure 5).

Thus, during the cambial activity period, the PCD of differentiating xylem cells occurs, which is under molecular genetic control, while these processes occur more intensely in the xylem derivatives of the younger cambium, compared to the xylem formed by the older cambium.

### 2.5. Expression of PCD Genes during the Transition from Sapwood to the Transition Zone

The transition from sapwood to the transition zone and heartwood characterized by the PCD of xylem ray parenchyma cells was accompanied by changes in PCD gene expression profiles in the transition zone compared to the inner sapwood [11,12]. In xylem ray parenchyma cells formed by a young cambium, cell death during the transition to the transition zone occurs faster than in cells formed by the older cambium. In this regard, it was interesting to study the comparison of the expression profiles of PCD genes during the transition from inner sapwood to the transition zone in sections of the trunk with different cambial ages.

Of all the genes studied, expression in the inner sapwood and transition zone was shown only for genes of the *BFN* family (*BFN*, *BFN1*, *BFN2*). The *BFN* gene had the highest expression levels. A significant increase in the expression level of the *BFN* and *BFN1* genes was shown during the transition from the inner sapwood to the transition zone, while a high negative correlation of the expression level of these genes with the cambial age in transition zone was observed (Figure 6). Pearson correlation coefficient was −0.88 (*p* < 0.01) for the *BFN* gene and −0.7 (*p* < 0.01) for the *BFN1* gene. For the inner sapwood, a significant correlation between the expression level and cambial age was shown only for the *BFN2* gene; on the contrary, in the transition zone, the expression level of this gene did not correlate with cambial age.

## 3. Discussion

### 3.1. PCD in Xylem Occurs during the Cambial Activity Period

The xylem (wood) of a tree trunk is a tissue complex comprising different cell types. All xylem cells have the same genesis: they are derived from the division of cambium cells. However, the functions performed in the xylem by cells of different types differ significantly, which causes differences in the program of their differentiation [16]. Pine xylem comprises tracheids, which perform water conducting and mechanical functions, and parenchyma cells, which form rays and a system of resin ducts. Tracheids undergo PCD soon after their formation and are dead cells in their mature state [17,18]. The formation of tracheids occurs during the cambial activity period and includes the stage of formation of mother xylem cells from cambium cells, elongation, formation of a secondary cell wall, and, as a final stage, PCD. Our measurements in the outer sapwood (differentiating xylem and xylem of current years) showed an increase in the expression level of the *BFN3*, *CEP*, and *MC5* genes in the cambial activity period compared to the cambial dormancy period. Previous studies have shown that these genes have tissue-specific expression during xylem differentiation in coniferous plants [12,19]. Xylem ray parenchyma cells remain alive for a long period, participating in many biological processes [10,20]. We showed that xylem ray parenchyma cells that reside at the sapwood/heartwood boundary in the transition zone also undergo PCD during the cambial activity period. Additionally, we showed the high expression of *BFN* and *BFN1* genes in the transition zone during the cambial activity period (Figure 2). In the cambial dormancy period, from October to January, the deposition of extractive substances occurs in the death zone of xylem ray parenchyma cells, while the transition zone narrows due to the transition from sapwood to heartwood [1]. Then, gradually over the course of a year, the tracheids lose water (transition from sapwood to the transition zone). In the cambial dormancy period, we showed a significant decrease in the expression level of the *BFN* and *BFN1* genes in the transition zone compared to the cambial activity period (Figure 2).

### 3.2. Expression of PCD Genes in the Xylem Differentiation Process Depends on the Cambial Age

In the outer sapwood, for most of the studied PCD genes, a negative correlation with cambial age was shown, i.e., it was shown that in outer sapwood cells formed by the younger cambium, the PCD gene expression was higher than in cells formed by the older cambium. It is known that cambial cells of woody plants maintain the ability to divide throughout life [21]. In older trees or in sections of the trunk with older cambium, the width of annual rings and the number of cells in the radial row of the xylem decreases [22,23], this trend can be explained by the fact that the size of a tree increases with age, the number of xylem cells around the circumference of the trunk increases, while these trees need fewer xylem cells to conduct the required amount of water to the tree crown [24]. More cells undergo PCD during the differentiation of xylem derivatives of the younger cambium, which is likely the reason for the increase in PCD gene expression in the stem regions with younger cambium.

### 3.3. Cambial Age and PCD of Xylem Ray Parenchyma Cells

The share of living xylem ray parenchyma cells will gradually decrease into the inner sapwood. Transcriptome analysis of the transition zone in *P. sylvestris* conducted by Lim et al. showed that the transition zone is largely represented by transcripts associated with secondary metabolism (pinosylvin biosynthesis), hormonal signaling, dehydration, and modification of the secondary cell wall (lignification) [11], while the main sign of the transition from sapwood to heartwood is the absence of nuclei in the cells of the ray parenchyma, i.e., their PCD. Analysis of principal components using the expression level of PCD genes as a data set made it possible to distinguish the transition zone into a separate group (Figure 3), which showed differences in the molecular genetic regulation of PCD during the differentiation of xylem derivatives of the cambium and during heartwood formation.

Previous studies have shown that PCD initiation of xylem ray parenchyma cells may occur due to their aging [25,26] and/or this process may be influenced by their position relative to the cambial zone [27,28]. Nagai et al., in their study using the example of *Cryptomeria japonica,* showed that the aging of xylem ray parenchyma cells is not directly related to their death and heartwood formation, since it was shown that the number of sapwood growth rings decreased with the trunk height, i.e., radial parenchyma cells formed by the younger cambium undergo PCD earlier compared to cells formed by the older cambium [28]. Our study also showed a decrease in the number of sapwood rings with the increasing trunk height, while the percentage of heartwood in the total trunk area did not change with height (Figure 1B). Measuring the radial distance from the heartwood boundary to the cambial zone also did not show significant differences in this indicator by height (Figure 1C), i.e., PCD initiation in xylem ray parenchyma cells in the trunk section with older and younger cambium occurred at approximately the same distance from the cambial zone. Nagai et al. concluded that the death of xylem ray parenchyma cells at a certain distance from the cambial zone may be caused by a lack of oxygen in the inner sapwood [28].

We showed an increase in the expression level of PCD genes as the tree trunk height increased and cambial age decreased. The reasons behind the detected changes remain unclear. We hypothesized that a higher level of gene expression may be associated with an increase in the number of cells undergoing PCD in the trunk section with younger cambium in relation to the older cambium; however, we showed that the transition zone width did not differ significantly across the entire trunk height (Figure 1C), i.e., it cannot have a significant effect on the PCD gene expression pattern. It is possible that the hormonal control and different distances from the tree top play a significant role in the expression regulation level of PCD genes

## 4. Materials and Methods

### 4.1. Study Objects

The study was conducted on 80-year-old Scots pine trees, depending on the cambial age under the lingonberry pine forest conditions in the transition area of the northern taiga subzone and tundra (Pasvik Nature Reserve, Murmansk region, Pechenga district). Five model trees were selected for the study. The dominant trees without oppression signs and damage were selected as model trees.

### 4.2. Plant Sampling

The trunk tissue samples were collected in the cambial activity period (28 June–3 July 2022) and cambial dormancy period (7–10 October 2021). Wood cores were collected with the Pressler borer (diameter 12 mm). Cores were obtained at different heights and divided into the outer sapwood (current year xylem and one annual ring of previous year xylem), inner sapwood (two annual rings before the transition zone), and transition zone according to anatomical research (Appendix A). Samples were immediately fixed for further investigation.

### 4.3. Microscopy

The wood core samples were immediately fixed in 70° ethanol solution. Fixed samples for microscopic analysis were delivered to the laboratory, placed in a refrigerator, and stored at +4 °C until further processing. Radial sections of wood with a thickness of 15–20 µm were prepared on a Tissue-Tek Cryo3 Flex freezing microtome (Sakura Finetek, Torrance, CA, USA). Sections were stained with a mixture of 1% alcohol solution of Alcian blue and 1% alcohol solution of safranin to identify the state of the pore membranes of tracheids and 4% solution of acetocarmine to identify nuclei [29]. Temporary preparations were made using glycerol as a mounting medium.

Photographing of sections was conducted using an Axio Imager A1 microscope (Carl Zeiss, Oberkochen, Germany) equipped with an ADF PRO 03 camera (ADF Optics, Wuhan, China). Photo processing was performed using ADF Image Capture software (ADF Optics, China). Using photographs of sections, the age of growth rings was determined, within which the deposition of extractive substances on the pore membranes of the bordered pores of tracheids and the disappearance of nuclei occurred. Heartwood was considered the part of xylem that lacked nuclei in parenchyma cells. The transition zone was considered to be a part of the wood in which the deposition of extractive substances was detected on the pore membranes of tracheids and nuclei in parenchyma cells were present (Appendix A).

### 4.4. qRT-PCR

Plant tissues for further qRT-PCR were immediately fixed in liquid nitrogen. In laboratory conditions, tissues were ground in liquid nitrogen and total RNA extraction was carried out by the CTAB buffer (pH 4.8–5.0). Then, 100 mM Tris-HCl (pH 8.0), 25 mM EDTA, 2M NaCl, 2% CTAB, 2% PVP, and 2% mercaptethanol were added to the mixture before use. Using a chloroform-isoamyl mixture (24:1), the mixture was separated into aqueous and organic phases. Total RNA was precipitated in the aqueous phase with absolute isopropanol. Next, to remove genomic DNA from the mixture, the total RNA with DNAse preparation (Syntol, Moscow, Russia) was incubated at 37 °C for an hour. Inactivation of DNAse was carried out by heating the mixture at 70 °C for 10 min before performing the reverse transcription reaction. RT reaction was performed using «T100 Thermalcycler» (BioRad, Foster City, CA, USA) with a set of MMLVRT reagents (Evrogen, Moscow, Russia) using Oligo(dT)15-and Random (dN)10-primer. The reaction mixture for PCR (25 µL totally) contained 5 µL qPCRmix-HS SYBR (Evrogen, Moscow, Russia), 1 µL of forward and reverse primers (0.4 µM) (Synthol, Moscow, Russia), 2 µL of template cDNA, and 16 µL of deionized, nuclease-free water. The final content of the cDNA reaction mixture for all samples was ~100 ng. qRT-PCR was performed under the following conditions: 95 °C for 5 min for a further 40 cycles, denaturation (95 °C, 15 s), annealing (52.7–61.6 °C, 30 s), and elongation (72 °C, 30 s). For each pair of primers, a negative control was used—PCR was performed in the absence of a cDNA template. The *GAPDH* gene was used as a reference gene for calculating the relative expression of genes, which, according to the analysis using BestKeeper and NormFinder, was the only gene stably expressed in all tissues studied. The primer sequence used for qRT-PCR is as follows: *GAPDH* forward primer (F) 5′*GGACAGTGGAAGCATCAT*3′, reverse primer (R) 5′*AACCGAATACAGCAACAGA*3′; *BFN*(F) 5′*GGCTTACAAAGACGCTGAGG*3′, *BFN*(R) 5′*CTGAATCCCGAGTGTGGTCT*3′; *BFN1*(F) 5′*CCATAATGCCGAAGGAGAA*3′, *BFN1*(R) 5′*GCTCTGCTGCCATAAGTT*3′; *BFN2*(F) 5′*AAGACGCTGATGAAGACA*3′, *BFN2*(R) 5′*CCAACCTTACACCTCCTT*3′; *BFN3*(F) 5′*TGATGAGATTCGTTATTG*3′, *BFN3*(R) 5′*CTGGTCAGTATAATTGTTA*3′; *CEP*(F) 5′*AAGGAATCAATTACTGG*ATAG3′, *CEP*(R) 5′*TTCAACTGCTTCAATACC*3′; *MC5*(F) 5′*TAACGCTCTTCAATCAAT*3′, *MC5*(R) 5′*ATGCTGTGAGTATTCTTC*3′.

The relative quantity of gene transcripts (RQ) was calculated from the formula:RQ = E^−ΔCt^,(1)
where ΔCt is the difference in the threshold cycle values for the reference and target genes, and E is the effectiveness of PCR. The effectiveness of PCR was determined individually for each reaction based on amplification fluorescence data using the LinRegPCR software (version 2021.1, Dr. J.M. Ruijter, Amsterdam UMC, Amsterdam, The Netherlands) [30].

### 4.5. Statistical Data Processing

The results were statistically processed with PAST (version 4.13). Before starting the statistical analysis, raw data were initially tested for normality using the Shapiro–Wilk test. The significance of differences between variants was estimated by the Mann–Whitney U test. The significant difference was evaluated at the level of *p* < 0.05. The analysis was carried out on five biological replicates. 

Principal component analysis and analysis of similarities (ANOSIM) was carried out for a data set for PCD genes expression in the trunk section with different cambial ages of five model trees. Before the calculations, the initial data were standardized.

## 5. Conclusions

The data obtained may provide additional information on regulating cambial activity in woody plants. Previous attempts to assess the influence of cambial age on various molecular genetic pathways did not provide the expected result, since it was shown that in Scots pine trees of different ages there is a shift in the timing of cambial activity, which plays a decisive role in determining the expression pattern of certain genes [23]. The existing variability of trees of different ages does not allow us to assess the influence of one factor (in particular, cambial age). The selection of material from different sections of the trunk of the same tree, with different cambial ages, makes it possible to neutralize the influence of soil and climatic conditions, as well as the phase of cambial growth and individual characteristics on the anatomical, morphological, and molecular genetic characteristics of trunk tissues. The approach we used allowed us to achieve an interesting result and reveal the possible influence of the height of tissue selection (cambial age) on the expression pattern of PCD genes during the formation of trunk tissues in *P. sylvestris*. Despite this, the mechanisms of this influence are not fully understood and require further research. Future work may be aimed at identifying specific genetic markers and key factors that determine these interactions.

## Figures and Tables

**Figure 1 plants-12-04072-f001:**
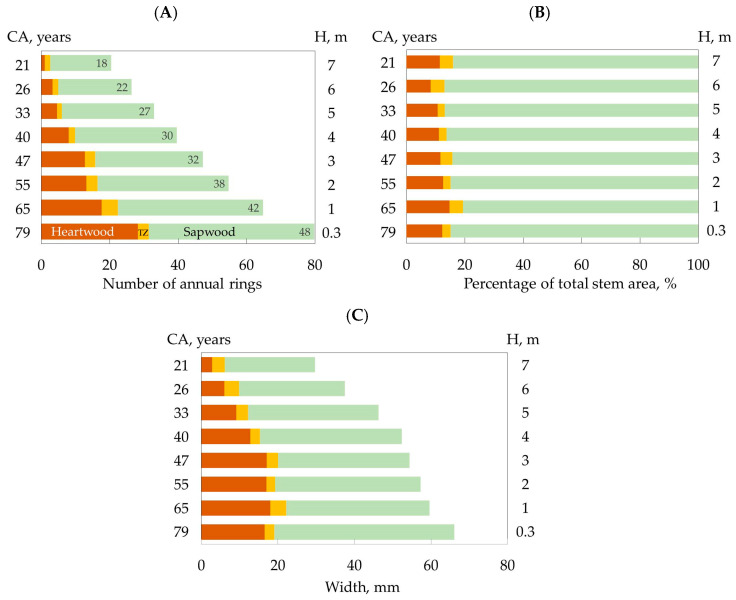
Number of annual rings of heartwood, sapwood, and transition zone (TZ) at different heights (H) of stem and cambial ages (CAs) (**A**); area percentage sapwood, transition zone, and heartwood of total stem area (%) (**B**); width of sapwood, heartwood and transition zone (mm) in 80-year-old Scots pine trees (**C**); values are averages of five biological replicates.

**Figure 2 plants-12-04072-f002:**
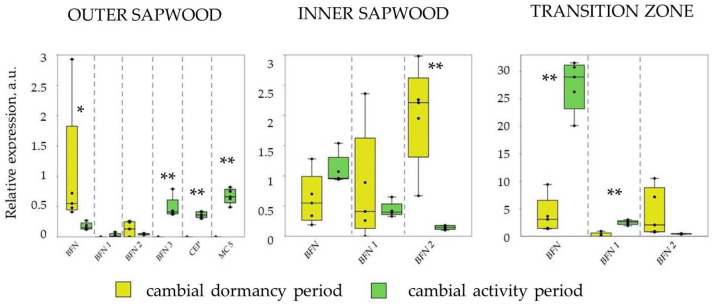
Expression level of PCD genes in different layers of xylem (outer sapwood, inner sapwood, and transition zone) of 80-year-old Scots pine trees in cambial dormancy (7–10 October 2021) and cambial activity period (28 June–3 July 2022) in the transition area of the northern taiga subzone and tundra (Pasvik Nature Reserve, Murmansk region, Pechenga district). Asterisks show differences in gene expression between cambial dormancy and cambial activity period. ******—significant differences at *p* < 0.01; *****—at *p* < 0.05. Relative expression level was calculated based on five biological replicates.

**Figure 3 plants-12-04072-f003:**
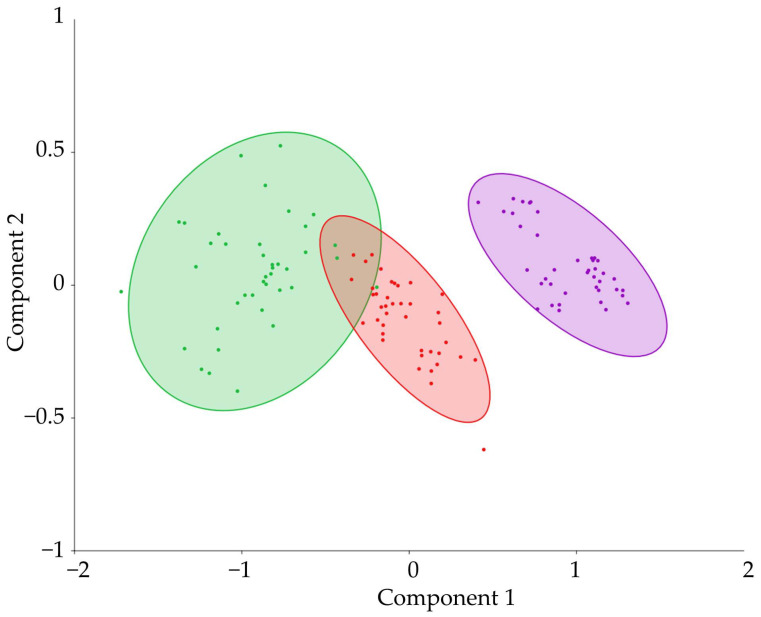
Principal component analysis based on gene expression of bifunctional endonuclease (*BFN)* genes family in trunk tissues at different heights (0.3, 1, 2, 3, 4, 5, 6, 7 m) of five model trees. Factor 1 (92% of the variance) was correlated with *BFN* (r = 0.64), *BFN1* (0.41), *BFN2* (0.64). Factor 2 (5% of the variance) was correlated with *BFN* (0.56), *BFN1* (0.32), and negatively correlated with *BFN2* (−0.76). The green grouping represents the outer sapwood. The red grouping represents the inner sapwood. The violet grouping represents the transition zone. The values shown are means with 95% confidence intervals.

**Figure 4 plants-12-04072-f004:**
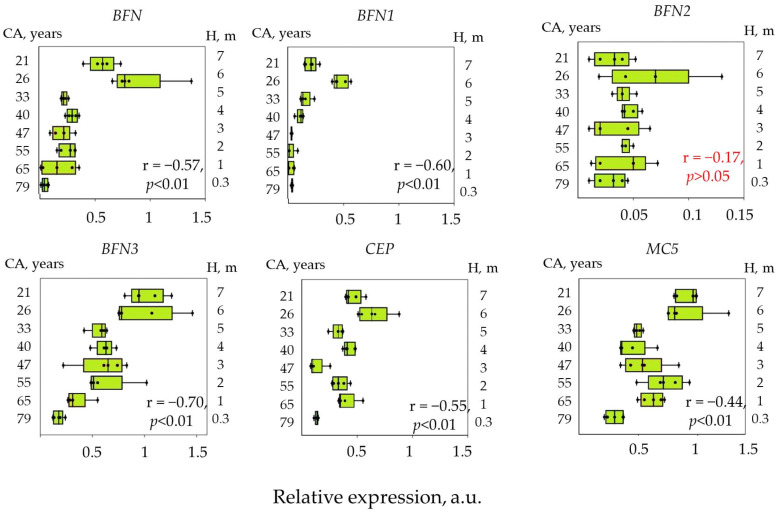
Relative expression of PCD genes in the outer sapwood of 80-year-old *Pinus sylvestris* L. trees into cambial age (CA) and height (H). Expression is calculated relative to the reference gene (*GAPDH*) and expressed in arbitrary units (a.u.). The graphs show the Pearson correlation coefficient (r). Dots indicate expression values in each model tree. Relative expression level was calculated based on five biological replicates.

**Figure 5 plants-12-04072-f005:**
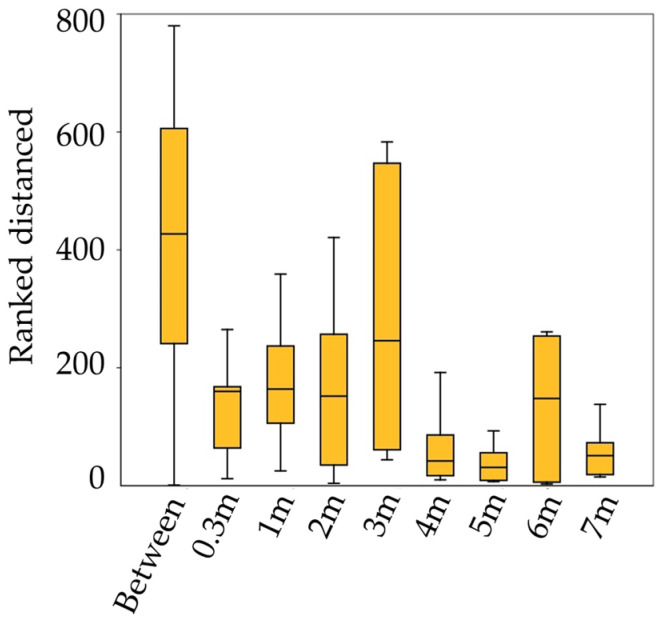
Ranked distance analysis of similarities (ANOSIM with Bray−Curtis similarity) to compare the sections on different trunk heights (permutation N = 9999, mean rank within = 125.6, mean rank between = 420.8, R = 0.7568, and *p* (same) = 0.0001).

**Figure 6 plants-12-04072-f006:**
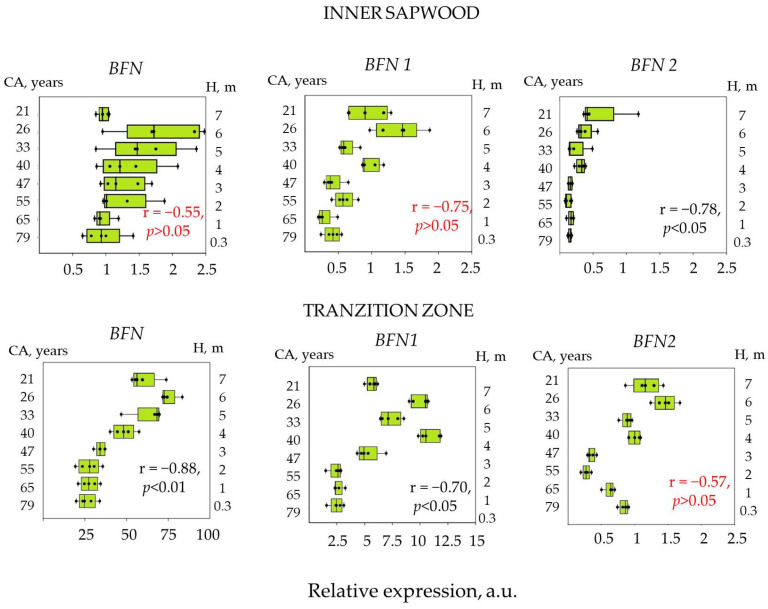
Relative expression level of PCD genes in the inner sapwood and the transition zone of 80-year-old Scots pine trees into cambial age (CA) and trunk height (H). The graph shows the Pearson correlation coefficient. Insignificant correlation is stated in red. Dots indicate expression values in each model tree Relative expression level was calculated based on five biological replicates.

## Data Availability

All data included in the main text and Appendix A.

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
