# Peer review of "Cambial Age Influences PCD Gene Expression during Xylem Development and Heartwood Formation"

_plants, 2023, doi:10.3390/plants12234072_

Round 1
Reviewer 1 Report
Comments and Suggestions for Authors
Very interesting research topic. Please see my comments in the attached document.

Comments on the Quality of English LanguageWell-written, in general. Only a few corrections needed.
Reviewer 2 Report
Comments and Suggestions for Authors
1. The part of introduction should be focus on the PCD or wood formation affected by cambium age, but I didn’t find that. And of course, what is PCD and what is general happened during this living processing was lacking.
2. How to ensure the right time for sample collection of cambial activity period and dormancy period? Did you consider that was varying in different years?
3. When in the title the heartwood formation was mentioned but why you didn’t collection samples from inner part of transition zone?
4. The most questionable thing is how do you prove the xylem tissue selected was during PCD vital activity? No morphological display.
5. What is the fixed condition of xylem tissue for qRT-PCR?
Comments on the Quality of English Languageno
Reviewer 3 Report
Comments and Suggestions for Authors
In general it’s a nice study. I wrote several remarks below.
Please remove the abbreviations- CA, HW, SW etc from the article. It’s very hard to remember all those, and a general reader will most certainly get lost in your article. Please use the entire phrases. Only PCD is a legitimate abbreviation, as it’s widely used instead of the entire phrase.
The Introduction lacks the information on the genes. It should prepare the reader for the results, but here we see the genes for the first time in the figures without any beforehand knowledge. For instance, what are the BFN genes? Please add.
Please add an image of an anatomical cross section showing all the tissues of the stem- cambial zone, heartwood, etc.
I could not understand the Figure 2 at all. There are many abbreviations and the legend is very unclear.
Figure 5 should be shown in the beginning, before the molecular data.
Please add information on biological repeats.
